# A qualitative inquiry on drivers of COVID-19 vaccine hesitancy among adults in Kenya

Stacey Orangi[1,2]*, Daniel Mbuthia[1], Elwyn Chondo[3], Carol Ngunu[4], Evelyn Kabia[1], John Ojal[5], Edwine Barasa[1,2,6]

1 Health Economics Research Unit, KEMRI-Wellcome Trust Research Program, Nairobi, Kenya, 2 Institute of Healthcare Management, Strathmore University, Nairobi, Kenya, 3 Department of Health, Kilifi, Kenya, 4 Nairobi City County Government, Nairobi, Kenya, 5 Kenya Medical Research Institute-Wellcome Trust Research Programme, Kilifi, Kenya, 6 Centre for Tropical Medicine and Global Health, Nuffield Department of Medicine, University of Oxford, Oxford, United Kingdom

* sorangi@kemri-wellcome.org

## Abstract

COVID-19 vaccination rates have been low among adults in Kenya (36.7% as of late March 2023) with vaccine hesitancy posing a threat to the COVID-19 vaccination program. This study sought to examine facilitators and barriers to COVID-19 vaccinations in Kenya. We conducted a qualitative cross-sectional study in two purposively selected counties in Kenya. We collected data through 8 focus group discussions with 80 community members and 8 in-depth interviews with health care managers and providers. The data was analyzed using a framework approach focusing on determinants of vaccine hesitancy and their influence on psychological constructs. Barriers to COVID-19 vaccine uptake were related to individual characteristics (males, younger age, perceived health status, belief in herbal medicine, and the lack of autonomy in decision making among women - especially in rural settings), contextual influences (lifting of bans, myths, medical mistrust, cultural and religious beliefs), and COVID-19 vaccine related factors (fear of unknown consequences, side-effects, lack of understanding on how vaccines work and rationale for boosters). However, community health volunteers, trusted leaders, mandates, financial and geographic access influenced COVID-19 vaccine uptake. These drivers of hesitancy mainly related to psychological constructs including confidence, complacency, and constraints. Vaccine hesitancy in Kenya is driven by multiple interconnected factors. These factors are likely to inform evidence-based targeted strategies that are built on trust to address vaccine hesitancy. These strategies could include gender responsive immunization programs, appropriate messaging and consistent communication that target fear, safety concerns, misconceptions and information gaps in line with community concerns. There is need to ensure that the strategies are tested in the local setting and incorporate a multisectoral approach including community health volunteers, religious leaders and community leaders.

**Data Availability Statement:** The data presented in this study are available upon reasonable request from the corresponding author through the email dgc@kemri-wellcome.org. Public deposition of the

transcripts would breach compliance with the approved protocol. During ethical approval, the transcripts were stated would be available publicly upon reasonable request through KEMRI-Wellcome Trust's data governance committee through the email dgc@kemri-wellcome.org. This was because at the time of data collection, it was deemed a sensitive topic, it was paramount not to compromise patient privacy, and allow free participation among the respondents. We are therefore kindly requesting for this exemption.

**Funding:** This work was supported by funding from the International Decision Support Initiative (IDSI) (EB, SO). Additional funds from a Wellcome Trust core grant awarded to the KEMRI-Wellcome Trust Research Program (#092654) (EB) and the German Academic Exchange Service (DAAD) (SO). The funders had no role in the study design, data collection and analysis, decision to publish, or preparation of the manuscript.

**Competing interests:** The authors have declared that no competing interests exist.

## Introduction

The Coronavirus disease (COVID-19) was first made a public health emergency in January 2020 and has led to more than 770 million confirmed cases and 7 million deaths globally (as of January 2024) [1]. In May 2023, the World Health Organization (WHO) declared that the COVID-19 disease was no longer a public health emergency [2]. However, there still remains the risk of new emerging variants that could result in a surge of cases and deaths [2].

Due to this risk, the importance of COVID-19 vaccines in protecting against serious illness, hospitalizations, and deaths, persists [3]. Kenya launched the rollout of the COVID-19 vaccines in March 2021 through a phased approach, with plans to vaccinate 100% of the adult population by December 2022. However in late March 2023, two years after the vaccine rollout in the country, only 36.7% of all adults had been fully vaccinated against COVID-19 with multi-vaccine types available in the country (including AstraZeneca, Pfizer, Johnson & Johnsons, Moderna, and Sinopharm vaccines) [4]. This low COVID-19 vaccine uptake could in part be attributed to inequities in access that led to delays in availability of COVID-19 vaccines in low-and middle-income countries [5, 6]. Nonetheless efforts were put in place to address these initial supply-side challenges in Kenya including procurement through the COVID-19 Vaccines Global Access Facility (COVAX), the African Union's African Vaccine Acquisition Task Team mechanism, and bilateral negotiations. Despite these efforts that ramped up vaccine supply, low vaccine uptake persists. This implies that demand side factors such as vaccine hesitancy threatens vaccine uptake, with estimates of more than a third of the Kenyan adult population classified as vaccine hesitant at the onset of vaccine rollout in the country [7].

Historically in Kenya, routine vaccinations have focused on children and adolescents through the Kenya Expanded Program for Immunization, making adult vaccinations a relatively new initiative, further emphasized by the COVID-19 pandemic. For instance, the seasonal influenza vaccine was recommended for introduction in Kenya in 2016 among children 6 to 23 months of age but was not recommended for any other risk group due to the lack of local burden of disease data [8].

Vaccine hesitancy is defined as the delay in acceptance or complete refusal of vaccination despite their availability [9]. Vaccine hesitancy is deemed to be complex, context specific, varies between time and vaccines and ranges on a continuum from overt acceptance to uncertainty, delay and outright refusal [10]. The prevalence of vaccine hesitancy over several decades and the threat it poses in reversing progress made in addressing vaccine preventable diseases has led to it being listed as one of the top 10 threats to global health in 2019 [11, 12]. Further, Kumar et al (2022) highlight that vaccine hesitancy during the pandemic could have been in multiple phases driven by societal reactions to vaccinations. These include 1) vaccination eagerness in the beginning to reduce mortalities, minimize lockdowns and resume normal life, 2) vaccination ignorance on the development process, safety, efficacy and appropriateness to the vaccine, 3) vaccination resistance led by anti-vaxxer movements, 4) vaccination confidence seen when morbidity and mortality due to COVID-19 was predominantly among the unvaccinated, 5) vaccination complacency preventing people from being fully vaccinated, and 6) vaccination apathy due to disinterest in vaccination [13].

Over the years, there have been different theoretical and conceptual frameworks that have been proposed to assess vaccine hesitancy. Initially focusing on childhood immunizations [10, 14–16] but increasingly focusing on COVID-19 vaccines [17–19]. In 2015, the 3Cs model categorized determinants of vaccine hesitancy across three categories: confidence, complacency, and convenience [10]. Later, the WHO Strategic Advisory Group of Experts on Immunization (SAGE) described vaccine hesitancy determinants matrix across three main categories: contextual influences, individual and group influences, and vaccine/vaccination specific issues [16].

In the recent years, there was the development of the 5C psychological antecedents that went beyond vaccine confidence and the system that delivers it to include confidence, complacency, constraints, calculation and collective responsibility [15]. A study looking into drivers of vaccine hesitancy in childhood vaccines in the African context used a conceptual framework incorporating three drivers of vaccine hesitancy i.e. caregiver-related factors, health systems related factors and community context [14, 15]. The framework assumed that these factors make individuals hesitant by influencing one or more of the 5C psychological constructs [14, 15]. The health belief model has also been used as a guide to explore factors influencing health beliefs regarding vaccine hesitancy [17, 18].

Globally, COVID-19 vaccine hesitancy across different demographic and cultural contexts has been reported to be influenced by risk perceptions, previous experiences with vaccines, trust in health care systems, misinformation, concerns about side effects and political ideology [20]. Further, low-income and middle-income countries report vaccine hesitancy among 20% of adults with safety and efficacy concerns being the most cited reason for hesitancy [21]. Previous evidence in Kenya highlight COVID-19 vaccine hesitancy as a challenge with almost all the studies using quantitative methods to determine drivers of vaccine hesitancy [7, 22–30]. Some of the studies focus on sub-groups of the Kenyan population including refugees, [27, 29] pregnant and lactating women, [23, 31] community health volunteers [24] and the youth [25]. Despite this, ongoing research into vaccine hesitancy is needed to understand the attitudes, beliefs and decisions around adult vaccination, in general, as this may have an impact on preparedness for future pandemics and future routine vaccines rolled out to adults such as the seasonal influenza vaccine. Specifically, there is need for more qualitative studies for in-depth exploration of drivers that influence people to be vaccinated or not, which might be helpful in shaping the future strategies of adult vaccination in the Kenyan context.

In this study, we explore participants' decisions around whether or not to receive the COVID-19 vaccine and the reasons behind them. By doing so, we are contributing to the understanding of facilitators and barriers to adult vaccinations in the Kenyan context.

## Methods

### Conceptual framework

We adapted a conceptual framework to assess the different determinants of vaccine hesitancy as derived largely from scoping reviews of literature on drivers of COVID-19 vaccine uptake [18, 19]. Moreover, given the cross-cutting themes with frameworks from childhood immunizations, we also incorporated relevant determinants from literature focussed on childhood immunization [10, 14–16]. Determinants of vaccine hesitancy were broadly classified across 4 sub-groups based on WHO SAGE framework [16]. Further, appropriate drivers of vaccine hesitancy identified from other frameworks were also classified into the 4 sub-groups [14, 17–19]. First, individual perceptions which explored how socio-demographic factors and individual perceptions such as past experiences with other vaccinations, trust in herbal medicine, conspiracy beliefs, perceived general health, altruism and collectivism influence vaccine hesitancy. Second, contextual and social influences, such as the communication and media environment, vaccine recommendations, trust in health systems/providers/government, religion/culture, and geographic barriers, and their influence on vaccine hesitancy. Third, COVID-19 vaccine and related factors including attitudes and beliefs, vaccine benefits/vaccine risks, knowledge and awareness on vaccination, length of development and clinical testing, and the type of vaccine. Lastly, COVID-19 infection and related factors such as perceived susceptibility and severity to COVID-19 (for self and others), and knowledge, attitudes, and practices regarding COVID-19 infection and their influence on vaccine hesitancy.

Given the criticism that vaccine hesitancy has previously neglected psychological states in its definition [32, 33], our conceptual framework also sought to analyse the influence of the aforementioned determinants of vaccine hesitancy on one-or more of the 5C psychological constructs: confidence, complacency, constraints, calculation, and collective responsibility [15]. Confidence refers to the trust in safety and effectiveness of the vaccine, the system that delivers it and the motivation of the leaders who decide on the need for vaccines [15]. Complacency exists when vaccination is not deemed necessary and there is a low perceived risk of the disease [15]. Constraints refers to structural and psychological barriers while, calculation refers to a person's engagement in information searching [15]. Lastly, collective responsibility is the willingness to protect others through herd immunity by being vaccinated [15].

In our conceptual framework, the four determinants of vaccine hesitancy interact, and influence one or more of the psychological constructs which in turn determine an individual's decision on vaccination. The individual decision about vaccination lies across a spectrum from complete refusal to complete acceptance of the vaccine and may vary across time. Decisions on COVID-19 vaccination that are either complete refusal, refusal but not certain (initially refuses and may delay in getting vaccinated), or acceptance but not certain (although accepted to be vaccinated, are uncertain about their decision) comprises vaccine hesitancy. This is illustrated in Fig 1.

The limitation of using a conceptual framework in data analysis is that the themes identified may be limited to those in line with the framework. However, to mitigate this, any new emerging themes outside of the conceptual framework were identified during analysis.

## Study setting

This study was conducted in two counties in Kenya, Nairobi and Kilifi. These counties were purposively selected to represent urban (Nairobi) and rural (Kilifi) settings, different geographic regions, and a high (Nairobi) versus low (Kilifi) COVID-19 vaccination coverage. This was done so as to identify any differences that may exist in the different settings. As of 22nd March 2023, Nairobi and Kilifi reported 57.1% and 21.3% of fully vaccinated adults respectively against the national average of 37.6% [4].

Focus group discussions (FGDs) and in-depth interviews (IDIs) were conducted in two sub-counties in Kilifi (Kilifi North and Malindi sub-counties) and Nairobi (Dagoretti and Makadara sub-counties). The sub-counties were purposively selected in consultation with county managers, to represent sub-counties with high and low vaccine coverage.

## Study design and data collection

We conducted a cross-sectional qualitative study design using 8 FGDs with 10 participants each and 8 IDIs with health care workers and county managers. FGDs were deemed an appropriate method to gain in-depth shared knowledge from community members on vaccine hesitancy. While IDIs were preferred to elicit individual views from health care workers on community member' vaccine hesitancy.

In each sub-county, two FGDs were done with the elderly (50 years and above) and those of a younger age (18–49 years). One or two community health volunteer(s) (CHV) were identified in each of the sub-counties. The CHVs assisted in purposive sampling and contacting eligible FGD participants from the community based on pre-specified varied characteristics such as socio-demographic parameters (age, gender, socio-economic, and education status) and COVID-19 vaccination status (a mix of those who were vaccinated, unvaccinated, and partially vaccinated). CHVs were deemed appropriate to contact FGD participants given their ability to

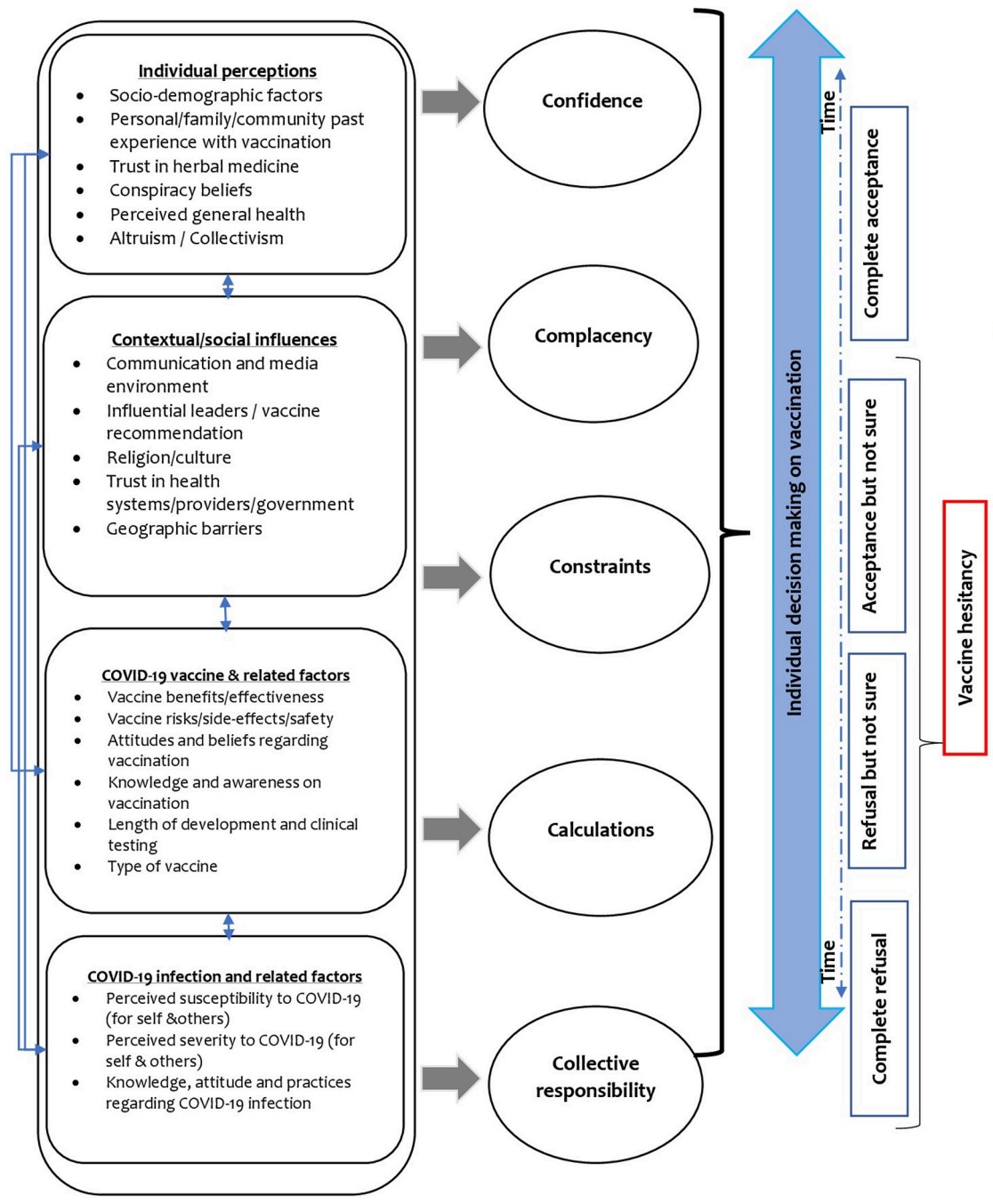

**Fig 1. Conceptual framework on drivers of vaccine hesitancy.**

easily identify community members based on the pre-specified criteria and their credibility in the community.

The participants were invited to a community hall where they had an initial information session to provide further information on the study and provide informed consent. The information session held by the researchers prior to the FGDs served to help mitigate the risk of undue influence that participants may have, through using CHVs in selecting potential

participants. FGDs were then conducted face-to-face in community halls for an average duration of 60 to 90 minutes. FGDs were conducted in Swahili, the language mainly used by community members in Nairobi and Kilifi. To supplement the FGDs, IDIs were conducted with health care workers and county managers involved in community demand generation activities of the COVID-19 vaccine. These IDI participants were purposively sampled and included 4 nurses, 1 community health assistant (identified from selected health centres and hospitals) and 3 county health managers involved in immunization activities. FGDs and IDIs were audio-recorded with the participants' consent and augmented by field notes. IDIs were conducted using semi-structured interview guides while discussion guides were used to moderate the FGDs. Questions in the interview and discussion guides were developed based on the study's conceptual framework (Fig 1) which assessed different determinants of vaccine hesitancy.

Debrief sessions during data collection were held between the two authors (SO, DM) to improve the interviewing process and aid with the revision of some questions in the interview/discussion guide for better clarity and to include emerging themes that needed to be explored further. Data collection was discontinued when data saturation was achieved. Data was collected from participants between 11 October 2022 and 14 December 2022.

## Participants

The FGDs participants comprised of both men (39 participants) and women (41 participants). The median age of the participants was 47.5 years with an age range of 18 years to 81 years. A majority of the participants were married (54%, n = 43), had a secondary school education (49%, n = 39), were unemployed (49%, n = 39), and were of Christian faith (88%, n = 70). Most of the FGD participants were vaccinated (80%, n = 64), with at least one dose and of those only about half (34 participants) had received the booster dose. 8 IDIs were conducted with health care workers involved in demand generation activities. Table 1 provides more details on the distribution of FGD participants in the study counties by socio-demographic characteristics.

## Data analysis

First, audio recordings were translated verbatim, those in Swahili were then translated to English. The data was then analysed using a framework approach that entailed familiarization, coding, charting, and interpreting the results [34]. We familiarized ourselves with the data by listening to the audios and re-reading the transcripts and field notes. The transcripts were then imported to NVIVO 12 (QSR International) for coding based on the determinants of vaccine hesitancy. Further, we mapped the influence (positive/negative) of the determinants on any of the psychological constructs and their influence on vaccine hesitancy illustrated through a causal loop diagram. In the casual loop diagram, how the variables influence each other is illustrated using an arrow. A feedback loop is shown when factors interconnect with each other. A balancing (B) feedback loop refers to the counter effect by the interconnected variables while a reinforcing (R) feedback loop refers to more of the effect being generated. After line-by-line coding and indexing all transcripts based on codes and themes, while ensuring all new emerging themes were captured, we charted the data. Charting the data into framework matrices was done by summarizing the themes and identifying illustrative quotes. Lastly, interpretation of the data was done by identifying connections between the various themes and gaining a better understanding on community members drivers of COVID-19 vaccine hesitancy.

**Table 1. FGD participant characteristics.**

| | Kilifi County | | | | | Nairobi County | | | | | Summary of FGD participants |
|---|---|---|---|---|---|---|---|---|---|---|---|
| | FGD 1 | FGD 2 | FGD 3 | FGD 4 | Total | FGD 5 | FGD 6 | FGD 7 | FGD 8 | Total | |
| **Sex** | | | | | | | | | | | |
| Male | 5 | 5 | 3 | 7 | **20** | 5 | 4 | 5 | 5 | **19** | 39 |
| Female | 5 | 5 | 7 | 3 | **20** | 5 | 6 | 5 | 5 | **20** | 41 |
| **Age** | | | | | | | | | | | |
| Median age | 26 | 66.5 | 56.5 | 31 | **49.5** | 30 | 67 | 28 | 63 | **46.5** | 47.5 |
| Age range | 18–44 | 50–79 | 50–73 | 19–49 | **18–79** | 19–47 | 45–81 | 19–48 | 50–74 | **19–81** | 18–81 |
| **Marital status** | | | | | | | | | | | |
| Married | 4 | 6 | 5 | 5 | **20** | 6 | 6 | 4 | 7 | **23** | 43 |
| Divorced/Separated | 1 | 1 | 2 | 1 | **5** | 0 | 1 | 0 | 0 | **1** | 6 |
| Widowed | 0 | 2 | 2 | 0 | **4** | 1 | 3 | 0 | 0 | **4** | 8 |
| Single | 5 | 0 | 1 | 4 | **10** | 3 | 0 | 6 | 3 | **12** | 22 |
| **Education** | | | | | | | | | | | |
| No education | 0 | 2 | 0 | 0 | **2** | 0 | 2 | 0 | 0 | **2** | 4 |
| Primary school | 2 | 4 | 2 | 2 | **10** | 1 | 4 | 0 | 6 | **11** | 21 |
| Secondary school | 5 | 4 | 7 | 5 | **21** | 6 | 3 | 7 | 2 | **18** | 39 |
| Tertiary school | 3 | 0 | 1 | 3 | **7** | 3 | 1 | 3 | 2 | **9** | 16 |
| **Employment** | | | | | | | | | | | |
| Full-time employment | 1 | 2 | 0 | 0 | **3** | 1 | 1 | 1 | 1 | **4** | 7 |
| Part-time employment | 2 | 0 | 3 | 3 | **8** | 1 | 0 | 1 | 3 | **5** | 13 |
| Self-employed | 1 | 3 | 1 | 2 | **7** | 2 | 4 | 0 | 2 | **8** | 15 |
| Student | 2 | 0 | 0 | 1 | **3** | 2 | 0 | 1 | 0 | **3** | 6 |
| Unemployed | 4 | 5 | 6 | 4 | **19** | 4 | 5 | 7 | 4 | **20** | 39 |
| **Religion** | | | | | | | | | | | |
| Christian | 8 | 7 | 7 | 8 | **30** | 10 | 10 | 10 | 10 | **40** | 70 |
| Muslim | 2 | 3 | 3 | 2 | **10** | 0 | 0 | 0 | 0 | **0** | 10 |
| **COVID-19 vaccination** | | | | | | | | | | | |
| Vaccinated | 8 | 9 | 10 | 9 | **36** | 5 | 9 | 9 | 5 | **28** | 64 |
| Unvaccinated | 2 | 1 | 0 | 1 | **4** | 5 | 1 | 1 | 5 | **12** | 16 |
| **Booster dose** | . | | | | | | | | | | |
| Received booster | 4 | 3 | 5 | 6 | **18** | 4 | 6 | 3 | 3 | **16** | 34 |
| Not received booster | 6 | 7 | 5 | 4 | **22** | 6 | 4 | 7 | 7 | **24** | 46 |

## Ethical considerations

Ethical approval for the study was obtained from the Kenya Medical Research Institute Scientific and Ethics Review Unit (KEMRI/SERU/CGMR-C 4244). Written informed consent was obtained from the participants prior to data collection. Participants were made aware that their participation was voluntary and that confidentiality in the research would be maintained.

## Results

The results are presented based on the four determinants of vaccine hesitancy: individual perceptions, contextual/social influences, COVID-19 vaccine & related factors, and COVID-19 infection and related factors. We also present how these determinants relate to any of the psychological constructs. Brief summaries of the key findings are provided under each determinant.

## Individual perceptions

**Differences in socio-demographic characteristics were drivers of vaccine hesitancy.**
Study respondents reported that those of a younger age were more hesitant to receive the
COVID-19 vaccine. This is because they had a perception of not being at risk of COVID-19.

*"Somebody is thinking they are currently healthy; maybe they are the ones who can go and
work. Most people say, "I am the breadwinner (but) you want to vaccinate me so that I'll be in
bed (due to side-effects)? Who's going to look after my family?"* **Nursing officer 3, rural
county**

Men compared to women reported to have poor health seeking behaviours and competing
work priorities that made them more hesitant to receive the vaccine. Further, the men per-
ceived that the COVID-19 vaccine had a side effect that affected their sexual health, and this
deterred them from vaccination.

*"Men refuse more than women. They refuse because they have a problem with going to hospi-
tal even if it is because of other diseases, other than corona . . .By the time they are going to
hospital the disease has spread and it's out of hand."* **FGD 6, urban county [over 50years]**

*"Men were the ones who had more issues and the youth from the rumours that they would
become impotent from the vaccines. Up to date, a lot of them still hold to that belief. "* **FGD7,
urban county [18-49years]**

In the rural areas, a majority of women lacked autonomy in deciding whether they should
be vaccinated. This responsibility lies with their husbands or partners. However, this did not
apply in the urban setting.

*"In most homes, women do not have the power to make decisions unless their husbands make
them. Even for the children, from the mother to the children. The man has to talk for us
(women) to accept. Until the man talks, they cannot agree to it. This is because he is the house
head, he has to know about it. This is because if you give a vaccine to the wife and she gets side
effects, it will be like the man failed in protecting the wife."* **FGD 3, rural county [18–49 years]**

*"I can say there is freedom of choice to get vaccinated for both men and women, because no
one was really asking the other if they should go get it, you just decide on your own and go."*
**FGD 5, urban county [18-49years]**

**There were beliefs in traditional or herbal remedies as alternatives to COVID-19 vacci-
nation.** In both the urban and rural settings, respondents cited community members having
beliefs in herbal medicines to prevent or treat COVID-19. In some cases, these remedies were
deemed sufficient and the need for vaccination was not perceived.

*"People in the interior areas have made their own drugs to prevent them from getting COVID.
They can tell you to take lemon, ginger and blend them together and consume and it will be
over. These lemons were not even available in the market . . .The people from tribe X also
have their beliefs, (some) do not believe in the hospitals. They believe in taking herbal medi-
cine."* **FGD 4, rural county [over 50 years]**

*"There was also another person saying that if you go to the market and buy lemon, boil water and put it with garlic, COVID will be so far from you, so we saw that there's no need of being vaccinated if you can treat yourself."* **FGD 8, urban county [over 50 years]**

**In some cases, health status and past experiences with vaccination were drivers of vaccine hesitancy.** While a majority of respondents reported that having comorbidities was a risk factor for COVID-19 infection, some respondents expressed concerns about being vaccinated while having comorbidities. Moreover, pregnant and lactating mothers were concerned about the safety of COVID-19 vaccines on the foetus and perceived that vaccination would cause reduced breast milk.

*"We were instilled fear. I have diabetes, high blood pressure and other conditions and we were made to fear that if a diabetic and hypertensive person goes to be injected they can die . . .I've not been injected because I have high blood pressure and feared."* **FGD 6, urban county [over 50years]**

*"Initially they (pregnant and lactating mothers) did not accept to it . . . I got my first dose when I was five months pregnant. There is another person who went for it after she saw that I have gotten it . . .They used to say that when you get it, the baby will die before being born or he will be disabled. Right now, people are enlightened, even those who are breastfeeding are getting it. Initially they used to say that when you get it when you are breastfeeding, you will not have enough milk to give your child."* **FGD 3, rural county [18-49years]**

Respondents past experiences with other vaccines (e.g. childhood vaccines) or other medicines administered through injections, influenced their uptake of COVID-19 vaccine. For example, respondents who had a fear of needles, were more likely to be hesitant to receive the COVID-19 vaccine.

*"I've been fearing vaccines since birth, not just COVID but any injection. I prefer being vaccinated in the ward so that I can't run away, so I just feared. I have not even been injected (COVID-19 vaccine), even one."* **FGD 8, urban county [over 50 years]**

*"There are other people who don't like injections. They are scared."* **FGD 1, rural county [18–49 years]**

## Contextual influences

**Structural differences in employment were a driver of vaccine hesitancy.** Respondents on a daily wage were reported to be more hesitant to receive the COVID-19 vaccine due to time constraints and the possibility of missing work due to side effects. This was unlike those in employment who have some protections such as paid sick leave.

*"We work on hand to mouth basis: you use up everything that you've made that day so I cannot get the time to go and get vaccinated."* **FGD 1, rural county [18-49years]**

**There were several reported media and communication channels which influenced perceptions of vaccines and as a result vaccine hesitancy/uptake to different extents.**

Respondents reported having heard about COVID-19 vaccination from different sources including international organizations like the World Health Organization, work, village leaders, media, health care workers, government and religious institutions. Although social media did play a role in vaccine uptake, it was also one of the greatest sources of promoting vaccine hesitancy.

*"Others got the information on COVID-19 from social media channels. When you enquire much about the information, you will find that some had exaggerated a lot. So, you will find that they have talked more of the negatives than the positives of COVID 19 vaccine. You will find young people or even people in the community are resisting because of what they have seen online." **FGD 3, rural county [18-49years]***

*"Again, social media: Facebook and Twitter, there was writing of negative stuff in relation to COVID vaccines. As you are trying to convince someone (to be vaccinated), they show you from their phones what is going on, and you see this thing (social media) has a large audience, so, they conclude this thing (COVID vaccine) is not good for human consumption." **FGD 7, urban county [18-49years]***

On the other hand, COVID-19 vaccines were taken up mostly due to information provided by CHVs. This is because CHVs are known and trusted by the community.

*"At first people did not believe that there is COVID, so through the CHVs and since they are close to the community many of us trust them . . .you find even there are other vaccines that people don't want to give their children but through the CHVs they trust them that there is no day they're going to introduce them to anything harmful." **FGD 1, rural county [18-49years]***

*"Here we do have CHVs, they were going around announcing and then later they set up vaccination stations. We believed them and went to get vaccinated. . .The CHVs, they are advising people and assuring those who are afraid. Like myself I was afraid but the CHV came and reassured me, and I got the vaccine." **FGD 5, urban county [18-49years]***

**Leaders were influential in promoting vaccine uptake and in some cases driving hesitancy.** Political and religious leaders either recommending the vaccine or taking the vaccine publicly was a key driver in vaccine uptake. However, some respondents reported a lack of trust in the government which was a driver of vaccine hesitancy.

*"We believed when we saw our president getting vaccinated. When I saw bishops who were above me being vaccinated, I saw it is for all and joined. " **FGD 8, urban county [over 50years]***

*"They also believe that the government is in business. When you go to people they will tell you that corona is over and the government is now doing business so as to get funding. . .Yes, they do not trust the government, what they say is not what they do." **FGD 4, rural county [over 50years]***

**There were cultural and religious beliefs that deterred vaccine uptake among community members.** In the rural setting, there were beliefs that a similar disease to COVID-19 existed in the past and was cured through the use of traditional medicine. Therefore, during the pandemic, the use of vaccines was not recognized.

*"There were others who were saying that this disease they are being told about now has been there since long time back. The people from tribe X called it 'kivuti'. Kivuti had its own medicine which people long time ago could take when they were sick . . .They were taking traditional medicines. They were questioning whether it is now that the others have learnt about the disease which has been there. They made traditional medicines which they believed treated them."* **FGD 4, rural county [over 50 years]**

*"When there was a 'kivuti' outbreak (in the past), traditional medicines were prepared, someone is given, and people are made to stay far from him. With time he would heal. Earlier on they had not discovered the medicine but later on they came to discover different trees which could be boiled and become a cure to the disease."* **FGD 2, rural county [over 50 years]**

Others reported having religious beliefs that made them hesitant to receive the vaccine.

*"There are those who do not believe in anything to do with medicine. When they get sick, they just kneel down and pray to God to heal them. If God does not heal them, then they are ready to die. There are many people who believe in that here."* **FGD 4, rural county [over 50years]**

*"For me, my mum told me to believe in God, same way He has kept me from those other diseases He will protect me from Corona too. So, for me, I just believe in God's protection over this."* **FGD 5, urban county [18-49years]**

**Non-pharmaceutical restrictions, movement restrictions, and other access restrictions that were enforced, promoted vaccine uptake.** Early on in the pandemic, there was enforcement of travel, work, and other access restrictions. These influenced some community members who were initially hesitant, to accept the COVID-19 vaccination.

*"I got it because the minister of health said you can't go to Mombasa if you have not been vaccinated so I just had to get vaccinated . . .Then also, there are those who want to travel to other countries so you find if you are not vaccinated you cannot cross the border or accepted in another country so that has an effect."* **FGD 1, rural county[18-49years]**

*"It was said that if someone is not vaccinated the person will not travel using a matatu, won't be paid, won't work, you see? So, I saw that my years (worked) will get lost. The years that I've served at work are many years, will they get lost because of three injections? I saw that I'd rather agree . . .if it were not for the rule that said no service will continue if you are not vaccinated, I would not have come."* **FGD 8, urban county [over 50years]**

However, removal of the travel restrictions, access restrictions, and other non-pharmaceutical interventions, coupled with the shifting government priorities created a perception that COVID-19 is no longer in existence or a threat hence there was no need for vaccination.

*"There are others who thought that since they have been told not to put on masks, there is no more corona. This affects people getting the vaccine."* **FGD 2, rural county [over 50years]**

*"Removal of the COVID measures is what caused the decline (in vaccination), everyone now thinks corona is gone . . .right now even if one received the second one or the booster, they don't see the need to go back, because it seems corona is gone . . .So, people were of the perception that now that the government has relaxed the measures it means the disease is gone."* **FGD 5, urban county [18-49years]**

**Geographic and financial barriers to access COVID-19 vaccines were addressed.** Initially, vaccination was only offered in health care facilities, resulting in long waiting times. However, the respondents reported that when the vaccination strategy evolved to a mixed approach including outreaches and mass campaigns, acceptance of COVID-19 vaccines increased. Transport costs and waiting times were no longer a barrier to access the vaccines. Highlighting ease of access being a driver of vaccine uptake.

*"You can get the vaccine anywhere, sometimes they go out for outreaches, door to door, sometimes schools, dispensary, churches, mosques . . . it has been good going to vaccinate people where they are, because most of them do not go to the dispensary to get these vaccines. When you take the vaccine next to them, when they see that others are getting them, they also get. Others wonder whether they have to go long distances, they request for the services to be brought near them. "* **FGD 3, rural county [18-49years]**

*"For me I didn't even go to look for that booster, it came to where I stay and I got injected. I see them coming there sometimes and when people hear that the injection is there as they pass, they go, get injected and continue with their journey."* **FGD 6, urban county [over 50years]**

**There were wide-spread rumours and myths that drove vaccine hesitancy.** Due to the novelty of the vaccine and novelty of vaccinating adults in Kenya, respondents reported having beliefs in several rumours and myths regarding COVID-19 vaccines. These included rumours that the vaccine would cause infertility, deaths among those vaccinated, and was being used as a tool to track and monitor people.

*"Many of them think it came to control the population. Now if you go to the community and ask "have you been vaccinated?" they say "wait for 10 years you're going to die". Others say you're going to turn into a zombie. Others say it's a form of tracking. Others are saying the population in Kenya is being reduced."* **FGD 1, rural county [18-49years]**

*" Even when the injection first came it was said that it belongs to people who are 58 and above so others said that they want to eradicate the elderly."* **FGD 6, urban county [over 50years]**

Despite these rumours, there were efforts to dispel the rumours through education and sensitization by the CHVs.

*"When COVID came around all CHVs were called, to get some training on what to teach the community. And from the information they had, they tried to give them the right information as opposed to the rumours."* **FGD 7, urban county [18-49years]**

## COVID-19 vaccine and related factors

**Although benefits of vaccination were recognized, the concerns on vaccine safety drove hesitancy.** Respondents cited the primary benefit of the COVID-19 vaccine as protection of self and others from COVID-19 infection and death. Other perceived benefits of vaccination by respondents included being able to work and travel without the risk of infection. However, as a result of the side effects that people were experiencing, there were concerns on the safety

of the vaccine. These side effects included pain on the injection site, fever, body aches, flu-like symptoms, among others.

*"On my part I have not been vaccinated. My husband was among the first to get the vaccine and when he came back after the vaccine, he really got sick . . . It took him like two weeks, being indoors sick and not going to work. He later on recovered. After some time, he went for the second jab, and still became sick . . . So, I have never been vaccinated, due to that scare."* **FGD 5, urban county [18-49years]**

*"People were scared of the second dose or the booster because of the side effects they got. The booster really made me sick . . . Like for 2 weeks, it really made me sick."* **FGD 1, rural county [18-49years]**

The short time taken to develop the vaccine was also a major concern to many respondents and raised questions on vaccine safety.

*"It was made very fast. They were saying that for a vaccine to be made, it needs to have been researched on for like ten years, but for this one that they have gotten within one year, they (community members) think that it is not safe."* **FGD 4, rural county [over 50years]**

*"We were more worried because we were telling each other that for HIV, (the vaccine) is yet to come around for all those years, how come the COVID-19 one got discovered that fast and COVID-19 kills people just like AIDS."* **FGD 7, urban county [18-49years]**

Respondents were also concerned about the unknown long-term side effects of the COVID-19 vaccine.

*"You know they have told us the immediate side-effects; we don't know about long term effects, if it is going to have long term effects or not. They should explain to us clearly the long-term effects rather than just the short-term effects. Maybe some organs might be affected, or other things might happen, so they need to be honest with us."* **FGD 1, rural county [18-49years]**

*"Fear of the unknown. A person can say that he does not know when he gets the vaccine, if he will get sicker."* **Nursing officer 2, rural county**

**There was a lack of knowledge and misinformation on vaccines which drove hesitancy.** There was misinformation among the respondents on how vaccines work in general. This coupled with a lack of understanding on the need for boosters, number of booster doses to be given, and the mixing of vaccine types, was a driver of COVID-19 vaccine hesitancy.

*"There are some who are told that you can get the virus from the vaccine, so the person sees it as if you're taking yourself to get COVID-19 . . .Others were saying, when they get sick, is when they'll be vaccinated, so, that when the vaccine goes in, it can fight off with the disease. They were adamant, they can't be vaccinated, and they are not sick, or in pain whatsoever."* **FGD 7, urban county [18-49years]**

*"People wonder that they have gotten a first dose from Moderna, Pfizer or Astra Zeneca, what is the need of a booster if the vaccine is good? Some will even question the booster, why get a*

*second booster? It is a challenge with those number of jabs. People are not comfortable with it.
" FGD 3, rural county [18-49years]*

**The availability of different vaccine types influenced vaccine acceptability.** There were vaccine preferences among the respondents that were largely based on the number of injections and experienced side effects.

*"Maybe someone saw somebody get AstraZeneca and got serious side effects when they come get vaccinated they say they don't want that. . .The government should make sure all the vaccines are available if someone comes and wants a certain vaccination it should be available. Someone may not get the vaccination they want and get demoralized." FGD 1, rural county [18-49years]*

*"Most people wanted that Johnson because it is one dose and you won't be pushed so much. You get injected once and won't go back to that injection again. Most people really wanted that one." FGD 6, urban county [over 50 years]*

Further, vaccine stockouts throughout the country of some of the vaccine types, influenced by supply-side challenges may be a driver of vaccine hesitancy. Especially to special populations groups that are eligible to receive certain vaccine types (e.g. pregnant women were eligible to receive mRNA vaccines) or those with specific vaccine preferences.

*"It (vaccine stockouts) affects me in a big way, especially, Pfizer. Pfizer is meant for pregnant mothers, lactating mothers, and children between 12 and 17. For instance, the last two months, Pfizer was out of stock in the whole country, so, we were not able to vaccinate those people, they were coming here, and we have to send them away. So, that's the biggest challenge when it comes to supply. I have these target populations I need to immunize, and I can't do it." Nursing officer 4, urban county*

**There were attitudes and beliefs around COVID-19 vaccination that drove hesitancy.** Respondents cited concerns on mixing vaccine brands and this would result in missing the second vaccine dose or not taking booster shots.

*"In the beginning when you were vaccinated with AstraZeneca you were to only be given AstraZeneca. For now it is not there. People wonder if they mix up the vaccines, it may affect them." FGD 1, rural county [18-49years]*

*"Another thing is that we fear the booster . . . Then you are told for the booster that you will not necessarily get the vaccine type that you got initially. You could get a different one. I am afraid of that up to now." FGD 2, rural county [over 50years]*

The novelty of adult vaccination in comparison to childhood vaccination was recognized as being a potential driver of vaccine hesitancy. Further, community members had adequate sensitization on the importance of childhood vaccines, unlike adult vaccines.

*"In my opinion all this is brought about because we are the first ones to use the COVID-19 vaccination but our next generation they are going to receive it well but you know when you're the first one to start something it is hard. The beginning of something is usually hard.*

*That is why we have these challenges . . . So maybe even when polio began it was hard for their parents." **FGD 1 rural county [18-49years]***

*"Children vaccination is very serious. Even when you don't have fare and you don't take them the father is going to quarrel you, "the child had a vaccination why didn't you go?" but now adult vaccination. . ." **FGD 1, rural county [18-49years]***

### COVID-19 infection and related factors

**Perceived susceptibility and likelihood of developing severe COVID-19 for self and others contributed to hesitancy.** A majority of the respondents perceived the elderly and those with co-morbidities to be the most susceptible to COVID-19 and therefore more likely to be vaccinated. The young had a lot of negative peer influence and perceived themselves to be of good health, hence driving hesitancy.

*"You know adolescents they are in groups. So, if one of them says they are not getting vaccinated the others will also follow. They say they're not going to get vaccinated. The old people are the ones to get COVID-19. So, it makes them say they have no need for that vaccination because they see they have a strong immunity and they're still young." **FGD 1, rural county [18-49years]***

### Influence on psychological constructs

The causal loop diagram (Fig 2) illustrates the linkage between the drivers of vaccine hesitancy, and how they are interrelated with the psychological constructs and vaccine hesitancy. The main psychological antecedents to vaccination were identified as confidence, constraints and complacency in this study setting.

The relationship between the three psychological constructs and vaccine hesitancy have reinforcing feedback loops as illustrated in the causal loop diagram. Therefore, when individuals have low confidence in the COVID-19 vaccine, are highly complacent, and there are high number of constraints in the system, this leads to increased vaccine hesitancy. Conversely, when vaccine hesitancy is high, individual's overall confidence in the vaccine is low, they would be more complacent, and there may be no motivation to address the constraints in the system.

## Discussion

This study reports qualitative findings on drivers of vaccine hesitancy and how they relate to psychological constructs among adults in Kenya, almost 2 years after vaccine rollout. This study suggests that drivers of vaccine hesitancy in Kenya, as in other settings are complex [20, 21], interrelated and a lack of confidence, high complacency and constraints are critical factors to consider for increased vaccine uptake.

In relation to individual characteristics, our study highlights that the youth were more complacent to receive the COVID-19 vaccine. This is because they report having a good health status and are more likely to interact with negative information on social media regarding the COVID-19 vaccine. Additionally, a lack of autonomy in decision making among women and children in rural settings was a constraint leading to vaccine hesitancy. This points out the

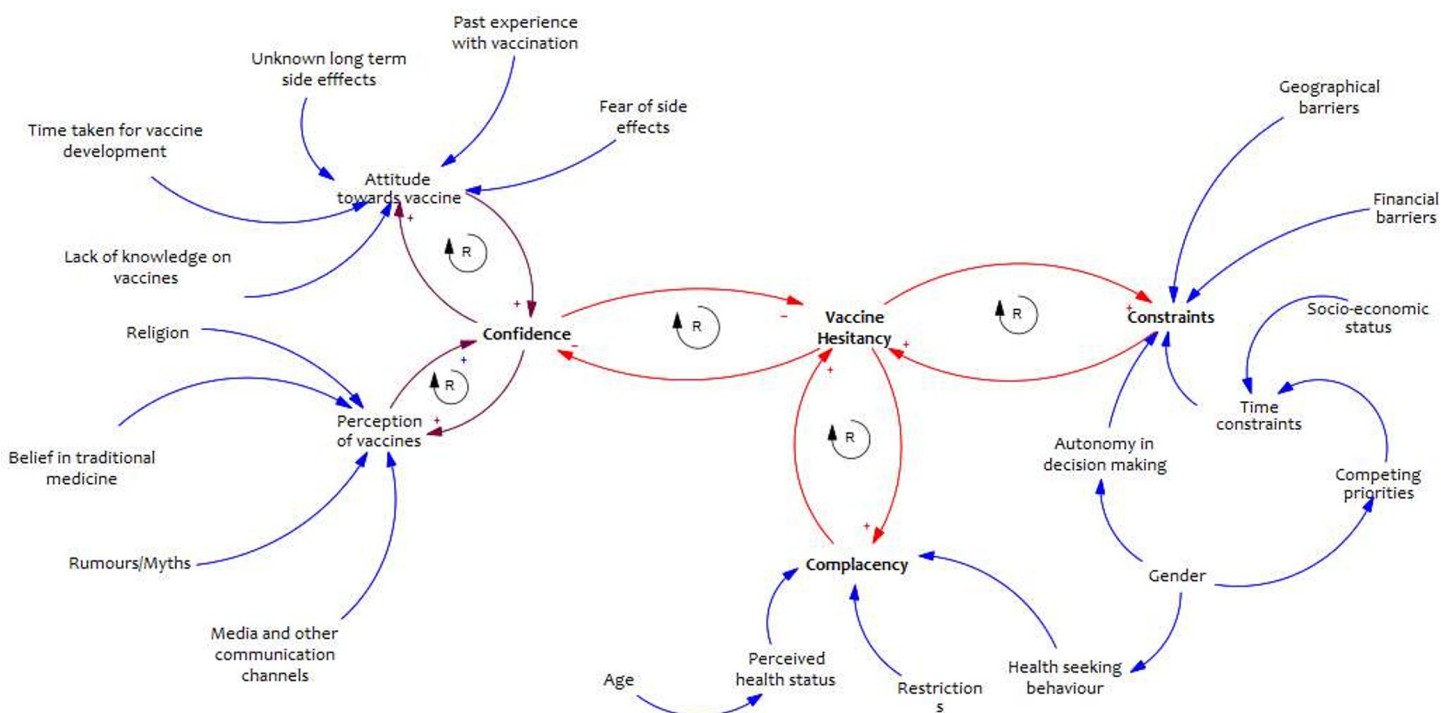

**Fig 2. Causal loop diagram of drivers and psychological constructs of COVID-19 vaccine hesitancy in Kenya.**

need for gender-responsive immunization programs that could include tailored community engagements with men, women, and the youth in the design of immunization delivery and implementation strategies while addressing their concerns [35]. Evidence from Kenya, highlights the need for more evidence-based engagements with the youth to increase uptake [25].

Several contextual factors were highlighted as drivers of vaccine hesitancy. First, the medium of communicating messages to promote COVID-19 vaccination is important in influencing vaccine confidence. There is consistent evidence from other studies in similar settings such as Cameroon, Ivory Coast and Nigeria that report social media platforms as a channel for inaccurate data on COVID-19 vaccines [36–38]. Social media's algorithm is tailored to compound exposure and reflect previous searches: Therefore if a person searches for and consumes information on vaccine hesitancy, they are more likely to be exposed to such content in future [39]. Further, there may be difficulty in a lay person determining credibility of the information [39]. Interestingly, experimental studies have demonstrated that viewing anti-vaccination content on social media increased negative beliefs on vaccination but viewing pro-vaccination content had minimal effects on beliefs [40, 41]. This highlights the need to identify the misinformation techniques used on social media (including conspiracies, fake experts, skewing the science, shifting hypotheses, censorship, misrepresentation and false logic), disentangle the core points and respond with evidence-based messages [42–44]. Further, there could be collaborations with technology platforms to spread accurate information that leverages on community-driven concerns.

In line with the findings of this study, CHVs who are trusted and perceived as influential should continuously be engaged, trained and incentivized to disseminate the evidence-based messages. This would also be instrumental in building health systems preparedness and resilience in future crises. Myths and rumours from this study that are likely to emerge during

other pandemics such as population control, long term negative effects can be used to inform preparedness for any future epidemics or pandemics.

Second, we report that religious and cultural beliefs also influence vaccine confidence and inform vaccine hesitancy. Similarly, a large majority from Niger, Liberia and Senegal report prayer as more effective in protection against COVID-19 compared to vaccination [45]. This emphasises the need for a multisectoral approach beyond the health sector and among various stakeholder groups. This deliberate collaboration and engagement of multiple sectors (such as with religious, cultural leaders, civil society) would leverage on diverse expertise, knowledge, reach and resources towards building trust in vaccines [46].

Third, lifting of restrictions and vaccination mandates led to individuals being complacent to vaccination. Our findings suggest that the enforcement of vaccination mandates for employment and travel compelled some community members who were hesitant to accept vaccination, because they valued their freedom to move and work. Further, upon removal of vaccination mandates, there was a perception that COVID-19 was no longer a risk, influencing vaccine uptake. There is evidence to suggest that COVID-19 vaccination mandates were associated with a rapid and significance rise in vaccinations in Canada and parts of Europe [47]. In Zimbabwe, there is report of high acceptance for COVID-19 vaccination mandates, which were strongly associated with perceptions of vaccine safety, effectiveness, and trust in the regulatory process [48]. On the contrary, vaccine mandates have also been reported to have unintended consequences including widening societal inequities, impacting on trust in governments, and reducing uptake of future public health measures [49, 50]. Further, vaccine mandates can impact on individual agency, a failure to fully address the root of vaccine hesitancy.

More sustainable approaches can be implemented that are built on trust and public debate to help the community to better understand the risk and benefits of the vaccine [49]. On the other hand, national mandates that allow for time away from work for vaccination during pandemics or work based vaccination programs could be considered to address productivity concerns and timings that were drivers to vaccine hesitancy.

COVID-19 vaccine safety concerns either because of the side effects experienced, the fear of unknown long-term side effects or the short time taken to develop the vaccine influenced respondents confidence in vaccination. Similarly, other evidence from Africa, South Asia, and Europe have highlighted concerns of vaccine safety, side effects and efficacy as one of the main concerns driving vaccine hesitancy [51–53]. While some experimental studies have shown the effect of different messaging for COVID-19 on vaccine intentions [54, 55], others report no effect on vaccine hesitancy by providing messages about vaccine risks and the development process to respondents [56]. This highlights the complexity in tailoring appropriate messaging, with appropriate frequency and delivery, especially in settings with slow vaccination coverage and a large vaccine hesitant population. Targeted messaging should be accompanied with message testing in the specific context as evidence suggests that messaging aimed at stimulating rational thinking of vaccine safety may in some cases not only be ineffective at positive change but could be counterproductive [57, 58]. Perhaps equally important, in addition to addressing COVID-19 vaccine safety concerns, our study highlights the need to provide a broader understanding on how vaccines work in general and the rationale for booster doses among the population, which would be beneficial in sensitisation to even other adult vaccines in the Kenyan context.

Our findings point out that vaccine hesitancy varies across time. Time-dimension coupled with exposure to new information and seeing others with positive experiences is a driver that can address hesitancy. This highlights the importance of consistent awareness building and right messaging over time.

In this study, we hypothesize reinforcing feedback loops between the three psychological constructs (confidence, complacency, and constraints) and vaccine hesitancy. Implying that when vaccine hesitancy is high, vaccine confidence is low, individuals are complacent and there are no motivations to address constrains in the system. On the other hand, when individuals have low confidence in the vaccine, are highly complacent and there are many constraints in the system, there is increased vaccine hesitancy. These feedback loops highlight the potential undesirable cycles that can consequently affect coverage level. Similar findings of reinforcing feedback loops between confidence and constraints with vaccine hesitancy are reported in a study from African settings on childhood vaccines [14].

We recommend further work on the relationship between the psychological constructs and determinants of hesitancy, especially for adult vaccines. There is also a need for future in-depth analysis on reasons why awareness campaigns in the local context may not have managed to dispel rumours and fears on the vaccine, as well as how traditional remedies are perceived.

The strength of this study is that it highlights the facilitators and barriers of COVID-19 vaccine hesitancy among community members (including both young adults and the elderly) in a rural and urban setting of Kenya. However, the study has some limitations. First, the findings of this study may not be generalizable, outside of these selected Kenyan counties. Second, due to the cross-sectional nature of the study and the fact that vaccine hesitancy varies across time, we were unable to account for temporal variations. Third, there may have been a social desirability bias among the participants to respond in a manner that is viewed favourable by others. This could have influenced how they self-reported their vaccination status or their narratives on vaccine hesitancy. Fourth, there was a lower participant consent rate among those unvaccinated at the time of the study.

This study aligns with initiatives promoting vaccine equity because beyond addressing inequitable vaccine distribution, overcoming vaccine hesitancy is necessary to achieve vaccine equity [59]. While COVID-19 is no longer a public health emergency, this study contributes to the knowledge of drivers of vaccine hesitancy in the local context that can inform preparedness of future pandemics and future routine vaccines rolled out to adults such as the seasonal influenza vaccine.

## Conclusions

COVID-19 vaccine hesitancy in Kenya is driven by multiple interconnected factors related to individual perceptions, contextual influences and COVID-19 vaccine and disease related concerns. These factors are likely to inform evidence-based targeted strategies that are built on trust to address vaccine hesitancy. These strategies could include gender responsive immunization programs, appropriate messaging and consistent communication that target fear, safety concerns, misconceptions and information gaps in line with community concerns. There is need to ensure that the strategies are tested in the local setting and incorporate a multisectoral approach including community health volunteers, community and religious leaders.

## Author Contributions

**Conceptualization:** Stacey Orangi, John Ojal, Edwine Barasa.

**Formal analysis:** Stacey Orangi.

**Funding acquisition:** Edwine Barasa.

**Investigation:** Stacey Orangi, Daniel Mbuthia.

**Methodology:** Stacey Orangi, Elwyn Chondo, Carol Ngunu, Evelyn Kabia, John Ojal, Edwine Barasa.

**Software:** Stacey Orangi.

**Supervision:** John Ojal, Edwine Barasa.

**Validation:** Daniel Mbuthia, Evelyn Kabia, Edwine Barasa.

**Writing – original draft:** Stacey Orangi.

**Writing – review & editing:** Stacey Orangi, Daniel Mbuthia, Elwyn Chondo, Carol Ngunu, Evelyn Kabia, John Ojal, Edwine Barasa.

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
