## [Decision Letter · Decision Letter 0]

7 Nov 2023

PGPH-D-23-02085

A qualitative inquiry on drivers of COVID-19 vaccine hesitancy among adults in Kenya

Dear Dr. Orangi,

Thank you for submitting your manuscript to PLOS Global Public Health. After careful consideration, we feel that it has merit but does not fully meet PLOS Global Public Health’s publication criteria as it currently stands. Therefore, we invite you to submit a revised version of the manuscript that addresses the points raised during the review process.

We look forward to receiving your revised manuscript.

Kind regards,

Lavanya Vijayasingham, PhD MPH

Academic Editor

Journal Requirements:

2. We have noticed that you have uploaded Supporting Information files, but you have not included a list of legends. Please add a full list of legends for your Supporting Information files after the references list.

3. In the online submission form, you indicated that "The data presented in this study are available upon reasonable request from the corresponding author". All PLOS journals now require all data underlying the findings described in their manuscript to be freely available to other researchers, either 1. In a public repository, 2. Within the manuscript itself, or 3. Uploaded as supplementary information.

Additional Editor Comments (if provided):

Thank you submitting this important research to this journal. We've had a speedy review process with meaningful comments and suggestions on how to enhance the manuscript.

I am recommending major revisions as the collective list of comments is long, and cover the whole manuscript. Nevertheless, I believe these points can be addressed well by the author team.

Overall please focus on:

a) the development and use of the conceptual framework- this appears to be an adaptation rather than a new or novel framework. Please also outline why the domains are independent, why these were necessary to distill from existing literature, how they differ in theory from the definitions of each of the 5Cs, and how they link to the 5Cs which are only discussed in the end of the results section and in the discussion. Please also attribute the components that came from prior models, and describe how

b) rationale for purposefully sampling such a low proportion of unvaccinated people- given the nature of the study and the national vaccinations rates presented. Please outline proportions who were partially vaccinated and their decision to not receive any further dose (not including booster shot- i.e 3rd dose of mRNA type of vaccines etc).

c) emergence of new themes or context specific knowledge that can contribute to tailored design and messaging of vaccination programs or any other health service delivery area, and how these findings are relevant especially since COVID-19 is no longer considered a public health emergency.

Specifically, please also consider and address the following points in the revisions (alongside those from the three reviewers). 

Introduction

1. Ln69 &70: “In Kenya, routinely vaccinating adults is a novel area, that has gained much appreciation since the COVID-19 pandemic”

What about seasonal Influenza? Perhaps elaborate on trends of access and uptake, including within high-risk populations- and reasons why adult vaccination was considered in the country as ‘novel’.

2. Th introduction section will benefit from outlining definitions and literature around adult vaccine hesitancy, contrasting this concept with vaccine acceptance (and discussion around the spectrum). Then discuss specifically for COVID-19 in global and LMIC context, before discussing the Kenyan context.

3. Which COVID-19 vaccines were available in Kenya during this time? What were specific themes of global hesitancy or concerns on each of these class/types of vaccines?

4. As the authors highlight- the COVID-19 is no longer a public health emergency. Please elaborate on why understanding the reasons for COVID-19 vaccine hesitancy is necessary now- perhaps in preparedness for future pandemics? Authors describe its impact on future routine vaccines in lines 85 &6- which adult vaccines?

Methods

1. Please provide rationale- why was a newly developed (or rather- an adapted version of the 5C psychological constructs) conceptual framework necessary? Is this an analytical framework instead?

2. Also, why were the four domains of determinants considered independent?

3. What about time dimension- did time change people’s perception and decisions? Even when they discussed the pandemic retrospectively?

4. How does the participant demographics in each setting reflect the composition in each of the geographical areas?

5. What was the reason for including vaccinated participants in a study about hesitancy- where 80% had at least one dose? How many had 2 doses (fully vaccinated)? Only 16 of all FGD were unvaccinated at time of FGD. Did any of those vaccinated even partially change their minds before first being hesitant, or after receiving the first dose? This speaks about the time component in the conceptual/analytical framework

6. What are characteristics of the 8 healthcare managers and providers?

7. In FGDs, what were observations around who spoke, and who were silent? What were group dynamics observed? How did the identity of the data collector likely influence engagement and discussions- especially tying in with social desirability which is discussed in limitations section.

Results

1. Why were results divided based pre-identified determinant from the scoping reviews, and how did the 5C model feature in the coding and analysis?

2. What were any new emergent codes and themes that fell outside the pre-established framework, and how do these provide new insights especially in the local context?

3. How did narratives from unvaccinated, partially vaccinated and fully vaccinated participants, including those Booster shots differ? How did the different dosing regimes influence decisions and narratives? Please include this data in the FGD participant table.

4. In relation to this- please also provide the proportion of vaccinated, partially vaccinated and fully vaccinated in each FGD group, and how this composition may have influence the discussions by groups. Did unvaccinated people or those chose to be partially vaccinated speak up during group discussion?

5. I think productivity concerns do not necessarily sit under differences in social demographic characteristics- it can be flipped around to be structural since people who work on daily wage will be missing out if they experience side effects or any longer term consequences in comparison to those with jobs that provide some ‘protections’ i.e- paid sick leave etc.

6. Gender, - gender roles and gendered patterns of health-seeking even outside vaccinations, is a notable dimension of uptake and hesitancy- not just a sociodemographic factor- but here biological sex plays a role too- in the quotes there is reference to ‘rumours that they would become impotent’ (ln 221)- this is a sex-related concern, where awareness campaigns have not managed to dispel these known fears. These areas require further in-depth analysis…

7. Myths, health status- the quote on lines 253-255 is an interesting one- that while this person may be seen as high risk of the negative consequences of infection, they are more fearful of the potential negative consequences of the vaccine.

8. The quote on lines 257 to 262 on use in pregnancy speaks about the time factor- that with time and exposure to new information and seeing other like them experience positively is a driver that can address hesitancy, and it also speaks to the importance of awareness building and right messaging.

9. In the introduction, the authors mention adult vaccinations being novel in Kenya, but speak about past experience of other vaccines in the uptake of COVID-19 vaccines, including fear of needles. Please elaborate on the types of adult vaccines that could have led to this, or alternatively with needles- could it also be any medicines administered through syringes?

10. Contextual- discussions in this section relate more to sociocultural influences than context: the role of social media and different sources of information, trusted information source/people- CHV-, religion, faith & traditional knowledge over modern medicines; Here the importance of capacity building and incentivizing CH workers or volunteers is useful to build health systems preparedness and resilience in future crises.

11. The discussion about Kivuti is interesting and is unique to the local context- what is the history of this, and is this discussed in medical anthropology literature?

12. Enforced restrictions as a driver that compelled people to be vaccinated- not necessarily addressed their hesitancy, but they valued their freedom to move, including for work- here another form of productivity concerns is presented. More discussions around policy enforcement as a structural driver is useful but discuss and comment on how it can impact on individual agency a failure to fully address the root of hesitancy. Vaccination is not the same as vaccine acceptance- so it will be important to discuss the influence of mandatory measures here.

13. Quote on line 365- speaks to vaccination programs and the venues or ease of access being a driver.

14. Myths and rumours- the different narratives around population control, long term negative effects and ‘eradication of elderly’ are interesting to prepare for any future epidemics/pandemics

15. The next section is similar- misinformation- how are these different to myths and rumours?

16. Availability of vaccines types is structural- what influenced this and what was available in these two settings? Any differences?

17. In the discussion around vaccine stock or stockouts- please explain why Pfizer was allocated to pregnant & lactating mothers, and why there were stock issues. “Sending them away” speaks to the convenience of access- where lack of stock, and structural factors that influence these barriers prevail.

18. Novelty of adult vaccination is context-based theme here: perhaps seasonal influenza shots are not common? Why? There is an interesting reference to polio by FGD participant (ln 466-467). Are there lessons that can be drawn from historical programs?

19. How were confidence, constraints and complacency identified as main antecedents? There is missing description and discussion of this.

20. Feedback loop and framework method- how did the analysis of themes contribute to the creation of the feedback loop? Please provide more details on process.

21. In the loop diagram, it appears that you are suggesting theoretical propositions or hypothesis that can be later tested. Is this the case? Please review and discuss how the relationships you propose features in current literature, and how it may be a contribution to research in this area.

Discussions

1. The results section is divided based on domains from scoping review but proposition, and now discussions is heavily based on 5Cs.

2. How are drivers of vaccine hesitancy any more complex in Kenya as opposed to any other country, or globally? (Ln 498)

3. Again, the link to lack of confidence, high complacency and constraints (Ln 499) is unclear.

4. The link between vaccine complacency and the lifting of structural restrictions is unclear based on the data and discussions presented in the results section. Again the need to discuss time is important- especially in how it influenced decisions or changing decisions.

5. The importance of gender-sensitive programs for vaccination, and those that address work factors i.e. timing, productivity concerns is important- perhaps a national mandate time away from work, or work based vaccination programs?

6. Consistency of language is useful- is the focus vaccine hesitancy or vaccine uptake? Ln 561

7. Limitations- why or what makes the study not generalizable? Indeed vaccine hesitancy varies across time (which further reiterates my point on the need to review the influence of time, or how decisions changed with time- as it is discussed retrospectively at the time of the data collection). How may have social desirability influenced the participants discussions or even the data that they self-report- i.e their vaccination status?

8. What specifically does the study contribute to knowledge of vaccine hesitancy in LMICs?

9. What are context or regional specific knowledge that requires further research? This could be how traditional remedies are perceived etc.

10. Again, while COVID-19 is no longer a public health emergency- what does this knowledge contribute to beyond COVID-18- perhaps in the future or in other health system areas?

Reviewers' comments:

Reviewer's Responses to Questions

**Comments to the Author**

1. Does this manuscript meet PLOS Global Public Health’s publication criteria? Is the manuscript technically sound, and do the data support the conclusions? The manuscript must describe methodologically and ethically rigorous research with conclusions that are appropriately drawn based on the data presented.

Reviewer #1: Yes

Reviewer #2: Yes

Reviewer #3: Partly

2. Has the statistical analysis been performed appropriately and rigorously?

Reviewer #1: N/A

Reviewer #2: N/A

Reviewer #3: Yes

3. Have the authors made all data underlying the findings in their manuscript fully available (please refer to the Data Availability Statement at the start of the manuscript PDF file)?

Reviewer #1: Yes

Reviewer #2: Yes

Reviewer #3: Yes

4. Is the manuscript presented in an intelligible fashion and written in standard English?

Reviewer #1: Yes

Reviewer #2: Yes

Reviewer #3: Yes

5. Review Comments to the Author

Reviewer #1: Thank you for the opportunity to review this important work. The area of work presented in this paper offers new and valuable insights into global health.

Feedback:

The conceptual framework referred to in the methods section should be introduced in the Introduction section. This is crucial to establish the main argument at the outset of the paper and to emphasize the primary findings discussed in the results section. This linkage between the introduction and the subsequent sections is currently missing and can be enhanced by conducting a more comprehensive review of the existing literature that forms the basis of the conceptual framework.

The results section is organized around the four determinants of vaccine hesitancy. However, the sub-headings (bolded) and the narrative within the results section sometimes diverge, and it would be beneficial to present the information more concisely and regroup it to clearly highlight the four determinants.

The discussion section would benefit from providing deeper insights into how this research aligns with global health initiatives promoting vaccine equity (based on the Results section) and discussing strategies for enhancing vaccine campaigns, especially across low- and middle-income countries (LMICs).

Reviewer #2: Introduction

The introduction successfully sets the global and local context of the COVID-19 pandemic, which provides a backdrop for the study's focus. A research gap in the existing literature is identified, underscoring the need for qualitative studies that offer in-depth insights into vaccination decisions.

In line 61, there's a goal mentioned about vaccinating 100% of the adult population by December 2022. Yet, by March 2023, only 36.7% of adults were vaccinated. It might be useful to question more explicitly why, aside from initial supply-side challenges, this ambitious target was missed so significantly.

The sentence in lines 69-71 regarding the novelty of adult vaccination in Kenya seems a bit indirect. Furthermore, the phrase '...has gained much appreciation…' seems to conflict with the subsequent introduction of vaccine hesitancy. Consider rephrasing for clarity: "Historically in Kenya, routine vaccinations targeted children and adolescents, making adult vaccination a relatively new initiative, further emphasized by the COVID-19 pandemic."

Line 73-75 discusses the definition and complexity of vaccine hesitancy. Consider briefly elaborating on why it was deemed one of the top 10 threats to global health in 2019 to provide readers with context.

In lines 77-79, Kumar et al.'s societal reactions to vaccinations might benefit from a brief explanation or example so readers can understand what each phase entails without referring to the original source.

Methods

The section on "Conceptual framework" offers a detailed breakdown of determinants of vaccine hesitancy. The 5C psychological constructs are well-defined. However, elaborating briefly on how these constructs were adapted specifically for this study may add depth.

In line 121, ’"they influence on or more of the psychological constructs" seems to have a typographical error.

While the spectrum from complete refusal to complete acceptance is mentioned, consider detailing more on the intermediate stages. What exactly constitutes "refusal but not certain" or "acceptance but not certain"? Providing examples or scenarios could enhance comprehension.

Towards the end, it might be useful to add a brief note about the limitations of the conceptual framework or any potential biases that could arise from its use.

Study settings

The section provides a succinct overview of the counties selected for the study. You can offer context as to why it was essential to study urban vs. rural, or high vs. low vaccination coverage.

The term "purposively selected" is used twice. Provide a brief rationale for the purposive selection method used, as readers may not be familiar with its application in this context.

Study design and data collection

Ensure consistency in terminology throughout, for example, ‘health care workers’ and ‘health workers’.

In line 149, the phrase "COVID-19 vaccination status" can be made clearer by specifying whether this refers to individuals who were vaccinated, unvaccinated, or partially vaccinated.

The role of community health volunteers (CHVs) is mentioned in lines 145-147. It would be beneficial to briefly explain why CHVs were chosen for this purpose and their credibility in the community.

The sentence in lines 166-168 repeats the idea of "saturation" twice, which can be redundant.

In Table 1, Ensure that the numbers in the "Summary of FGD participants" match the totals from Kilifi County and Nairobi County.

Results

The organization of results into four determinants of vaccine hesitancy is commendable. This makes it easier to follow and understand the various factors influencing vaccine hesitancy.

Addressing perceptions around comorbidities, pregnancy, and past experiences with vaccinations is relevant in the context of the COVID-19 pandemic. These factors have been topical issues in vaccine uptake discussions globally.

It might be beneficial to contrast the perceptions from urban vs. rural settings more explicitly. This would highlight the unique challenges and drivers in each setting.

The information on cultural and religious beliefs influencing vaccine hesitancy is intriguing. Delving deeper into this topic and providing more details would enhance understanding.

There's a bit of repetition in lines 331-332, where "restrictions" is repeated multiple times; it can be phrased concisely.

In line 356, “COVID-19vaccines” is missing a space.

In lines 483/488, the term "casual loop diagram" appears to be a typographical error.

In line 490, consider changing "then vaccine hesitancy will be high" to "this leads to increased vaccine hesitancy."

While many challenges and concerns are highlighted, there's limited mention of potential solutions or ways the concerns have been addressed, other than CHV efforts. Expanding on measures taken by health officials, NGOs, or local communities would provide a more balanced view.

Discussion

The discussion is logically structured with a focus on vaccine hesitancy's key drivers, the role of communication mediums, the impact of mandates, and constraints. The study context and its relevance two years post-vaccine rollout is adequately established.

The parallels drawn with similar studies from other regions strengthen the argument and contribute to the paper's depth. Mentioning social media's role in shaping vaccine perceptions is crucial in today's context, and this section effectively highlights its impact and the challenge of misinformation.

There's a typographical error in the beginning: "qualitative fi1ndings"

Consider expanding on the "multisectoral approach." While the inclusion of community health workers and religious leaders is mentioned, further elaboration on how these sectors can collaborate might enhance the recommendations' clarity.

Given the importance of combating misinformation on social media, consider suggesting specific strategies or collaborations with tech platforms or leveraging community-driven content to spread accurate information.

Conclusion

The conclusion is concise and to the point, effectively summarizing the discussion's main points. Consider expanding on the "transparent and consistent communication" recommendation, providing tangible steps or examples.

Reviewer #3: 1. What was the rationale of interviewing healthcare managers and providers, as the objective is to determine "decisions around whether or not to receive COVID-19 vaccine and the reasons behind them", as indicated in the introduction. The data seems hardly used as almost all excerpts were from FGDs.

2. In the discussion, the author should discuss impact of starting off their qualitative data analysis using a conceptual framework as opposed to letting the data "speak". Was a mixed approach utilized?

3. The authors purposively selected Kilifi and Nairobi county, based only on vaccination coverage. Given vaccination is largely contextual, and Kenya has 47 diverse counties, do the views represent vaccine hesitancy in Kenya?

4. The distribution of the 8 IDIS is broadly defined as health workers and county managers. Would good to specify which cadres and in what quantity of the 8 respondents given the demographics table is not included. Line 156/7.

5. In the introduction the authors reported that Kenya had a vaccination rate of 36.7%, and specifically indicate that selected counties had vaccination rates of 57.1% and 21.3%. However, the results indicate that 80% of the respondents were vaccinated, with a good majority getting the booster dose. Despite this being a qualitative study, collecting views on vaccination hesitancy should aim to balance those with the outcome, and at the least should have purposively selected based on vaccination status to at least a 50:50 ration to get feedback from majority who did not get vaccinated. This sample is the voice of largely vaccinated people. What dot he unvaccinated (majority in the country and selected counties) have to say? This voice is under represented. Is it the effect of using HCW to recruit study participants?

6. The findings indicate that there was a gender influence on vaccine uptake, especially in the rural area selected, driven by socio-cultural factors. However, it is noted that all FGDs were mixed gender as opposed to gender specific. Do the authors feel that they have accurately captured the weaker gender's perceptional and drivers to vaccination? What safeguards were in place to ensure free and open expression?

7. In the discussion, kindly be sure to tie it to your findings which you seem to have used various lenses. Started off with the conceptual framework with 4 key constructs, once the themes were developed they were further classified based on the 5C model. Hence the discussion should align with whichever framework or model the author would prefer to present, or clearly present both, so that the content is easier to digest.

6. PLOS authors have the option to publish the peer review history of their article (what does this mean?). If published, this will include your full peer review and any attached files.

**Do you want your identity to be public for this peer review?** For information about this choice, including consent withdrawal, please see our Privacy Policy.

Reviewer #1: No

Reviewer #2: No

Reviewer #3: No

---

## [Decision Letter · Decision Letter 1]

7 Mar 2024

A qualitative inquiry on drivers of COVID-19 vaccine hesitancy among adults in Kenya

PGPH-D-23-02085R1

Dear Ms Orangi,

We are pleased to inform you that your manuscript 'A qualitative inquiry on drivers of COVID-19 vaccine hesitancy among adults in Kenya' has been provisionally accepted for publication in PLOS Global Public Health.

Best regards,

Lavanya Vijayasingham, PhD MPH

Academic Editor

Thank you for comprehensively addressing the comments and suggestions provided in the previous round of reviews. The reviewers and I find that the manuscript has been refined to a high standard and are happy to accept it for publication.

Please also submit a COREQ checklist (as supporting material) when finalizing for publication. Congratulations on completing this important research, and wishing you all the best for future research endeavors.

Reviewer Comments (if any, and for reference):

Reviewer's Responses to Questions

**Comments to the Author**

1. If the authors have adequately addressed your comments raised in a previous round of review and you feel that this manuscript is now acceptable for publication, you may indicate that here to bypass the “Comments to the Author” section, enter your conflict of interest statement in the “Confidential to Editor” section, and submit your "Accept" recommendation.

Reviewer #2: All comments have been addressed

Reviewer #3: All comments have been addressed

2. Does this manuscript meet PLOS Global Public Health’s publication criteria? Is the manuscript technically sound, and do the data support the conclusions? The manuscript must describe methodologically and ethically rigorous research with conclusions that are appropriately drawn based on the data presented.

Reviewer #2: Yes

Reviewer #3: Yes

3. Has the statistical analysis been performed appropriately and rigorously?

Reviewer #2: N/A

Reviewer #3: Yes

4. Have the authors made all data underlying the findings in their manuscript fully available (please refer to the Data Availability Statement at the start of the manuscript PDF file)?

Reviewer #2: Yes

Reviewer #3: Yes

5. Is the manuscript presented in an intelligible fashion and written in standard English?

Reviewer #2: Yes

Reviewer #3: Yes

6. Review Comments to the Author

Reviewer #2: The manuscript on COVID-19 vaccination rates and hesitancy in Kenya has successfully addressed the detailed comments raised in the previous round of review and is now acceptable for publication in PLOS Global Public Health. The authors have conducted methodologically and ethically rigorous research, presenting their findings in a clearly and coherently, supported by appropriately analyzed interview data. The conclusions are well-founded, offering nuanced insights into the facilitators and barriers to COVID-19 vaccinations within the Kenyan context. The manuscript is technically sound, with a framework analysis that rigorously examines the determinants of vaccine hesitancy and their impact on psychological constructs. It is written in standard English, ensuring that the study is accessible to a broad audience. The authors propose evidence-based, targeted strategies to address vaccine hesitancy, highlighting the importance of trust-building, gender-responsive programs, and a multisectoral approach. This work significantly contributes to our understanding of vaccine hesitancy in Kenya and presents actionable recommendations for increasing vaccination uptake.

Reviewer #3: 1. Correct sentence structure on line 59/60. It does not read like a complete sentence.

7. PLOS authors have the option to publish the peer review history of their article (what does this mean?). If published, this will include your full peer review and any attached files.

**Do you want your identity to be public for this peer review?** For information about this choice, including consent withdrawal, please see our Privacy Policy.

Reviewer #2: **Yes: **Shazmin Khalid

Reviewer #3: No
